# MEG Node Degree Differences in Patients with Focal Epilepsy vs. Controls—Influence of Experimental Conditions

**DOI:** 10.3390/brainsci11121590

**Published:** 2021-11-30

**Authors:** Stephan Vogel, Martin Kaltenhäuser, Cora Kim, Nadia Müller-Voggel, Karl Rössler, Arnd Dörfler, Stefan Schwab, Hajo Hamer, Michael Buchfelder, Stefan Rampp

**Affiliations:** 1Department of Neurosurgery, University Hospital Erlangen, 91054 Erlangen, Germany; martin.kaltenhaeuser@uk-erlangen.de (M.K.); cora.kim@uk-erlangen.de (C.K.); nadia.mueller-voggel@uk-erlangen.de (N.M.-V.); michael.buchfelder@uk-erlangen.de (M.B.); stefan.rampp@uk-erlangen.de (S.R.); 2Friedrich Alexander University Erlangen Nürnberg (FAU), 91054 Erlangen, Germany; 3Department of Neurosurgery, Medical University Vienna, 1090 Vienna, Austria; karl.roessler@meduniwien.ac.at; 4Department of Neuroradiology, University Hospital Erlangen, 91054 Erlangen, Germany; arnd.doerfler@uk-erlangen.de; 5Department of Neurology, University Hospital Erlangen, 91054 Erlangen, Germany; stefan.schwab@uk-erlangen.de (S.S.); hajo.hamer@uk-erlangen.de (H.H.); 6Department of Neurosurgery, University Hospital Halle (Saale), 06120 Halle (Saale), Germany

**Keywords:** epilepsy, epilepsy surgery, epileptogenic focus localization, magnetencephalography, connectivity, node degree

## Abstract

Drug-resistant epilepsy can be most limiting for patients, and surgery represents a viable therapy option. With the growing research on the human connectome and the evidence of epilepsy being a network disorder, connectivity analysis may be able to contribute to our understanding of epilepsy and may be potentially developed into clinical applications. In this magnetoencephalographic study, we determined the whole-brain node degree of connectivity levels in patients and controls. Resting-state activity was measured at five frequency bands in 15 healthy controls and 15 patients with focal epilepsy of different etiologies. The whole-brain all-to-all imaginary part of coherence in source space was then calculated. Node degree was determined and parcellated and was used for further statistical evaluation. In comparison to controls, we found a significantly higher overall node degree in patients with lesional and non-lesional epilepsy. Furthermore, we examined the conditions of high/reduced vigilance and open/closed eyes in controls, to analyze whether patient node degree levels can be achieved. We evaluated intraclass-correlation statistics (ICC) to evaluate the reproducibility. Connectivity and specifically node degree analysis could present new tools for one of the most common neurological diseases, with potential applications in epilepsy diagnostics.

## 1. Introduction

As one of the most common neurological diseases, epilepsy is affecting over 50 million people worldwide [1]. A recent meta-analysis suggests new anti-epileptic drugs have not been able to treat sufficiently up to 25% of all patients [2], while persisting seizures or side effects of medication have a negative influence on each individual’s quality of life. This is where epilepsy surgery provides an alternative for long-term seizure control in such pharmacoresistant cases [3,4]. Depending on etiology, seizure freedom rates range from approximately 50% up to 90% [5,6,7,8], although recurrence after 2–5 years occurs in approx. 50% of patients [9,10]. The prerequisite for successful surgery and long-term seizure freedom is the exact localization and complete resection of the epileptogenic zone [11]. A broad spectrum of diagnostic modalities is applied to achieve this goal, ranging from EEG, MEG (electro-/magnetoencephalography), to MRI as well as SPECT and PET [12].

Among these, magnetoencephalography (MEG) has been shown to add valuable information to the presurgical epilepsy workup [13,14]. Specifically, it has been demonstrated that taking MEG source analysis findings into account for the planning of invasive recordings [15] and surgical resection results in higher postsurgical seizure freedom rates, even in the long term [16].

However, there are problems that remain. Routine analysis relies mainly on the evaluation of interictal epileptic discharges (IED). Correspondingly, if no IED occur during the comparably short recording periods, as is the case in approximately 20–30% of cases [16,17,18], the investigation does not yield any information on focus localization. In addition, interictal epileptic activity is thought to arise from the irritative zone, which may or may not overlap or be identical with the epileptogenic zone [11].

Furthermore, conventional source analysis of epileptic activity in MEG data is a time-intensive and user-dependent process, requiring specific expertise [19]. In the current study, we instead consider a connectome-based solution for the detection of epilepsy-related functional alterations.

The concept [20,21,22] is based on the idea of epilepsy as a network disorder [23,24]. It assumes that while epileptic activity may be generated locally within the epileptogenic or irritative zone, it is communicated to other, possibly distant, areas via pre-existing pathways. It is noteworthy that such epileptic activity might be subtle, variable and not necessarily correspond to visually identifiable patterns [20,25,26]. Generating areas would then appear as nodes within the network, with many incoming and outgoing functional connections. In graph theory, this characteristic is described by the so-called “node degree”, a measure counting the absolute number of such connecting edges of a network node [27]. Recent data suggests that overall functional connectivity in patients with both focal and generalized epilepsies is significantly increased compared to healthy controls [20,21,28,29,30]. Putatively, focus localization would then be conceivable by discovering nodes with exceedingly high node degrees or “hubs”. The current study focuses on this necessary prerequisite for any clinical application, i.e., whether node degree is truly increased in patients.

For such clinical applications, a range of experimental factors could potentially impact the recording and interpretation of connectivity and specifically of node degree analysis. It is well known that, for example, vigilance affects resting-state recordings in both fMRI and EEG tests [31]. Correspondingly, drowsiness during longer resting-state MEG recordings, as used for epileptic focus localization, may well impact any connectivity analysis. Furthermore, resting-state recordings may be conducted with the eyes open (EO) or eyes closed (EC), which may affect not only power but also connectivity characteristics in the alpha frequency band [32]. The variability and reproducibility of node degree in healthy controls could limit the reliability of exceedingly high node degree values in patients [33]. Finally, connectivity analysis can be applied separately to different frequency bands [22]; however, the optimal choice for node degree analysis remains unclear.

Recent studies have shown that EEG and MEG should be considered complementary not only for the conventional source imaging of epileptic activity [34] but also when investigating connectivity patterns [35,36]. These results demonstrate that both modalities provide distinct perspectives on resting-state dynamics, affected not only by different sensitivities but also by different susceptibility to volume conduction effects. Hypothetically, a combined analysis, yet again similar to IED analysis [37], should be able to combine the best of both worlds and yield a comprehensive view of connectivity.

In the presented study, we focus on MEG and evaluate the parameters of functional connectivity as a marker to distinguish patients with focal epilepsy from healthy controls. We specifically investigate if measurement conditions—vigilance and eyes open/closed—may lead to connectivity characteristics in healthy controls that abolish this contrast. Our findings provide novel insights on functional connectivity, which may add to the necessary foundation for implementing clinical applications and are also relevant for the use of connectivity measures in neurocognitive studies.

## 2. Materials and Methods

### 2.1. Participants

Both healthy controls and patients with pharmacoresistant focal epilepsies, undergoing presurgical evaluation, were investigated. All participants gave written informed consent for their participation in the study. The procedures of the study were reviewed and approved by the Ethics Committee of the Department of Medicine, University of Erlangen-Nuremberg (registration number 179_19B and 52_17B).

### 2.2. Controls

Fifteen healthy controls were included. All of them were recruited from social media or have already been part of other MEG studies (9 females, 6 males, mean age 26 years). The inclusion criteria were as follows: (1) physically and mentally healthy adults; (2) inconspicuous magnet-resonance imaging (MRI) of the neurocranium; and (3) less than 0.5 cm of detected movement in the MEG recording. The exclusion criteria were: (1) any long-term medication—except for contraceptives; (2) excessive movement during the MRI or MEG scan; and (3) metallic implants causing artifacts in the MEG recording/MRI scan.

### 2.3. Patients

We aimed at including 15 recent consecutive patients with pharmacoresistant focal epilepsies who were undergoing presurgical evaluation at the Epilepsy Center of the Department of Neurology, University Hospital Erlangen, Germany in 2019/2020. The inclusion criteria were: (1) adult patients with (2) focal- and (3) drug-resistant epilepsy [38]; (4) of lesional or cryptogenic etiology. The exclusion criteria were: (1) previous brain surgery/epilepsy surgery; (2) excessive movement during the MRI scan or MEG recording; and (3) metallic implants causing artifacts in the MEG recording/MRI scan. Initially, 18 consecutive patients were selected, of whom three had to be excluded for various reasons (Table 1). The remaining group consisted of 8 female and 7 male patients (mean age 34 years, in a range from 23 to 60).

### 2.4. MRI Scan

All participants (controls and patients) were scanned using a high-resolution 1.5 or 3T MRI-System (Siemens Magnetom Trio and Aera, Department of Neuroradiology, University Hospital of Erlangen) to acquire a T1-weighted, isotropic, 1 mm, 3D MP-RAGE (Siemens; magnetization prepared—rapid acquisition gradient echo) dataset.

### 2.5. MEG Recording

All MEG data were recorded with a 248-channel whole-head-system (WHS) (Magnes 3600 WH; 4D-Neuroimaging, San Diego, CA, USA) in a magnetically and electrically shielded room (Vacuumschmelze (VAC) GmbH, Hanau, HE, Germany).

Prior to data acquisition, participants were clothed in non-metallic outfits and were instructed to remove any make-up to avoid any artificial induction of magnetic fields. For head positioning in relation to the MEG sensors, five small coils were attached to the left and right pre- and retroauricular points and at the hairline. Additional head shape acquisition was performed using a 3D digitizing pen, covering the nose, the eyebrows, the attached coils and the cranium.

Both the healthy controls and the patients were arranged in a supine position within the WHS. Several recordings, named “runs”, were acquired with a duration of 5 to 20 min each, with sampling rates of 508 Hz or 678.2 Hz and a 1–339.1 Hz or 1–600 Hz analog filter. If head movement during a run exceeded >5 mm, the run was disregarded and repeated.

Patients were recorded once, for a total of 40 min duration and EC condition, as part of their presurgical diagnostics.

The controls underwent four recordings and were instructed not to move, think of anything in particular, or fall asleep during the experiment. The first recording was four minutes of resting state with EO. Participants were told to focus on a monitor projecting a cross. Second, we selected four minutes of resting state with EC, hereinafter referred to as “Start” and “EC”.

Afterward, an auditory listening task of 40 min duration had to be performed, intended to cause fatigue in our volunteers. During this recording, 140 tinnitus-like sounds of different frequencies were presented to the controls. The sounds varied slightly in their continuity and were paired with a background noise, making it harder to distinguish between a continuous and an interrupted tone. The task included rating each sound with a keyboard in four different levels of continuity, ranging from interrupted to continuous. As the volunteers had to focus for 40 min on slightly different tones, all agreed that this task was mentally exhausting and physically tiring in a follow-up survey.

Subsequently, another four minutes of continuous resting-state data were selected, hereinafter referred to as “End”.

### 2.6. MEG Data Analysis

All preprocessing and analysis steps were performed using field-trip software version 20200406 (https://www.fieldtriptoolbox.org, accessed on 20 October 2021) and SPM8 (https://www.fil.ion.ucl.ac.uk/spm/, accessed on 20 October 2021) on Matlab R2020b (The Mathworks, Natick, MA, USA).

#### 2.6.1. Pre-Processing

Brain and skin compartments were segmented using the individual MRI datasets (SPM8). MRI and MEG coordinate systems were co-registered by matching the acquired head-shape points to the segmented MRI skin surface. The segmented brain compartment was used to set up a single-shell volume conductor [39] and to fit an MNI-template grid with 10 mm resolution to the individual anatomy of each subject. The latter was performed to allow straightforward interindividual comparisons and calculations.

The first 4 min of the recordings were segmented into 2-second epochs. As later analysis steps focused on individual frequency bands, no digital filtering was applied. All epochs were then visually checked for artifacts and noisy channels, which were excluded from further processing. From the remainder, epochs with a total duration of 4 min were selected for analysis.

The resulting epochs were then down-sampled to 300 Hz. An independent component analysis (ICA) was calculated using the “runica” algorithm [40].

Components reflecting eye blink, eye movement and ECG artifacts were visually identified and removed. Epochs with IEDs were not discarded; seizures did not occur during the recordings.

#### 2.6.2. Source Analysis

The preprocessed data were subjected to a multitaper frequency transformation in 1 Hz steps in the frequency band of interest, with a DPSS taper and 2 Hz frequency smoothing: delta (1–4 Hz), theta (4–7 Hz), alpha (8–15 Hz), beta (15–30 Hz) and low gamma (30–45 Hz). Source analysis was then performed via the partial canonical coherence (PCC) method, using standard 5% regularization. Finally, the source-reconstructed data was reduced to the dominant orientation.

#### 2.6.3. Connectivity and Graph Analysis

The imaginary part of coherency (IMPC) was calculated between all source space-node pairs [39]. The resulting connectivity values were thresholded at the 95% percentile, to set up a corresponding graph. Node degree was then calculated for each source space node.

Finally, the node degree values were parcellated, based on the automatic anatomical labeling (AAL) atlas [41], calculating the maximum value over all the source space nodes of a region. The mean over all atlas regions, excluding the cerebellum and basal ganglia, was then taken as the subject mean for further statistical analysis.

#### 2.6.4. Power Analysis

Putatively, elevated node degree levels could be an epiphenomenon of similarly increased power in the respective frequency bands.

To investigate this aspect, we performed analytical and statistical evaluations using power values instead of node degree on the eyes-closed data. Specifically, for each source space node, PCC yielded projected power and noise values. An evaluation was then performed on the NAI-values (neural activity index) by dividing each power value by the respective noise estimate.

### 2.7. Statistics

Statistical evaluation was aimed at evaluating the contrast between the node degree values of patients vs. controls under different conditions. The underlying rationale was that a robust group difference would be advantageous for practical clinical application. We specifically investigated three specific aspects: reproducibility and the systematic differences of node degree in controls under different conditions, as well as the impact on the contrast to patients.

Reproducibility between conditions was evaluated using intraclass correlations (ICC) with a two-way mixed-effects model with single measurement, absolute agreement [42]. ICCs were calculated from within-subject node degree maxima over all parcels, as this parameter is used for the patient-control contrast.

Systematic differences between conditions were investigated with separate non-parametric cluster-permutation tests, using 1000 randomizations and a cluster alpha of 0.025 [43].

For the evaluation of the impact on patient-control contrast, node degree values were transformed to z-scores using the values observed in the healthy controls as a reference. Group differences were then evaluated by calculating a Wilcoxon rank-sum test. Furthermore, the ability to use node degree subject means as a marker to differentiate patients from controls enabled receiver-operator-characteristic (ROC) analysis to be performed, yielding area-under-the-curve values (AUC). Briefly, AUC values vary between 0 and 1. An AUC value of 1 designates perfect differentiation between the groups, whereas 0.5 would correspond to random chance. Values between 0 and 0.5 suggest that the label variable (i.e., patient vs. control) and the evaluated parameter values show an inverse relationship.

In our case, this would mean that the assumption that higher node degree values are observed in patients would be wrong and that, instead, the opposite should be assumed. Conventionally, AUC values between 0.9 and 1 are considered to reflect an excellent ability to differentiate, 0.8–0.9 is good, 0.7–0.8 is fair, and poor is between 0.6 and 0.7 [44].

Comparisons between patients and controls were calculated between maximum node degree (maxND) values for each condition, respectively recording in the controls, and eyes-open vs. eyes-closed at the beginning, rather than at the end, of the recording. Since only one recording was available for patients, these comparisons always utilized the EC condition for the patient group, corresponding to a clinical routine in our institution.

Finally, maxND per patient in this EC condition was correlated (Spearman correlation) to the IED rate (Table 1) determined during clinical routine from the complete, approx. 40-min, recording, from which the data for the ND-analysis was taken. This step served to evaluate a potential association between IED and ND, which may suggest that ND would be an epiphenomenon of IEDs.

## 3. Results

### 3.1. Eyes-Opened Compared to Eyes-Closed

15 healthy subjects had two successive recordings in a supine position, one with EO and one with EC. The resulting maxNDs were evaluated by comparing the conditions “EC—EO” for each parcel. We found no significant systematic node degree differences in any frequency band.

### 3.2. Eyes-Closed before and after a Demanding Task

15 healthy subjects were recorded before and after an auditory task with EC and in a supine position. The resulting maxNDs were evaluated by comparing the conditions “Start—End” for each parcel. We found no significant differences in any of the evaluated frequency bands.

### 3.3. ICC Evaluation of Control Data

Intra-class-correlation (ICC) statistics between different experimental conditions in our healthy control group were evaluated. The ICC of maximal node degree (Max. ND ICC) was calculated for the conditions “EO/EC” and “Start/End”. Reproducibility in healthy controls EO compared to EC condition did not exceed 0.53 (Delta—Control Open/Close, see Table 2) and showed less reliable results in other frequency bands (see Table 2).

Comparing the first run (Start) with the last (End) resulted in a maximal node degree ICC (Max. ND ICC) of 0.36 (Theta—Start/End, see Table 2), with other frequency bands being less reliable (Table 2).

### 3.4. Healthy Controls Compared to Patients with Focal Epilepsy

We compared data from the 15 healthy controls with our epilepsy group of 15 consecutive patients (Table 1). Versus healthy controls, we found a significantly increased overall node degree in the patient group (non-parametric Wilcoxon rank-sum test) within all frequency bands (Table 3 and Table 4/Figure 1 and Figure 2).

### 3.5. Node Degree Comparison between Healthy Controls at Different Experimental Conditions and Patients

With overall node degree being significantly increased in patients versus controls, we additionally compared node degree under different experimental conditions, measured in controls with EC patient data to verify reproducibility.

Comparing each of the two vigilance conditions (“Start” and “End”) to patient data suggests a higher node degree in epilepsy patients and appears to be significantly increased in all chosen frequency bands (AUC—Table 3/Figure 1).

Similar results can be visualized in EO/EC condition compared to patient data. EO or EC condition does not abolish the significant contrast between the patient and control group in any of the chosen frequency bands, with the exception of EO low gamma (*p* = 0.074). (AUC—Table 4/Figure 2.)

### 3.6. Comparison of Power between Patients and Controls

Power was compared between patients and controls in the EC condition (Table 5/Figure 3). Results showed no significant power difference in the alpha, beta and low gamma bands. Delta yielded a difference at the level of a tendency, whereas theta reached statistical significance.

### 3.7. Correlation of maxND and IED Rate

Spearman correlation between IED rate and maxND in the EC condition yielded non-significant correlations in all frequency bands before correction for multiple comparisons: r = 0.32 (*p* = 0.26) for delta, r = 0.36 (*p* = 0.19) for theta, r = 0.31 (*p* = 0.26) for alpha, r = −0.14 (*p* = 0.60) for beta and r = 0.20 (*p* = 0.47) for gamma.

## 4. Discussion

Our study investigates the robustness of MEG graph-theoretical node degree over different measurement conditions, specifically, the conditional impact on the contrast between patients with epilepsy and healthy controls. We found generally low reproducibility between conditions (median ICC of 0.3 over all frequency bands and conditions) in controls, with no observable systematic differences.

Despite this considerable variability, the contrast with patients with epilepsy was not affected. Independent from the measurement condition for comparison with controls, patient results always showed a significantly increased overall node degree.

### 4.1. Increased Node Degree as a Correlate of Epilepsy

The rationale of evaluating node degree as one of many graph-theoretical measures [45,46,47] is the hyperexcitability encountered in epilepsies. This characteristic feature leads to spontaneous, largely random interictal discharges, ranging from typical spikes and sharp waves to oscillatory patterns. In focal epilepsies, the spatial distribution of these is constrained to one or a few circumscribed areas. However, due to the interconnected nature of the brain’s architecture, we assume that this activation is communicated to connected, possibly distant areas. This may take the form of classical propagation but may also be more subtle and appear only as statistical connectivity. The node degree would then identify hubs within such a network, potentially corresponding to the source of spontaneous activity and subsequent functional connectivity [48].

In the current study, we focused on quantitative differences between patients with focal epilepsy and healthy controls. We did indeed observe significant contrasts in all evaluated frequency bands, which were not explained by the rate of IEDs in the investigated patients.

These results are concordant with those of Focke et al. (2018) who also investigated node degree; however, that study was in patients with focal and generalized epilepsy [21]. They also reported a significant increase in both groups in comparison to healthy controls. It is noteworthy that they investigated only patients without a lesion with an MRI. It seems, therefore, unlikely that the observed node degree increases are a purely lesional effect, e.g., similar to slowing in EEG in patients after intracranial surgery or with large lesions [49,50].

Further MEG/EEG studies report similar findings but evaluate increased functional connectivity rather than graph theoretical measures. An EEG study performed by Bettus et al. in 2008, for example, was one of the first to show increased functional connectivity in 21 patients with temporal lobe epilepsy [28], while a MEG study performed by Wu et al. (2014) identified abnormal imaginary coherence in epilepsy patients without quantitative measurements in 100% of their 30 patients [29].

Another MEG study by Elshahabi et al. (2015) found a widespread increase of connectivity in a collective of 13 patients with genetic generalized epilepsy vs. 23 controls [30].

However, Englot et al. (2015) have found contradicting results [17]. They report an overall decrease in connectivity in a group of presurgical focal epilepsy patients, compared to healthy controls. In fact, regionally reduced connectivity as a consequence of deleterious effects within the cortex could sharpen the contrast to the epilepsy-related overactive nodes, although this has yet to be confirmed. The selection of patients could be a further explanation, since our consecutive group included a majority of non-lesional patients (12 cryptogenic and 3 lesional) while their cohort mostly consisted of MRI-positive (51 lesional and 10 cryptogenic) patients. Differences in etiologies may also play a role, since gliomas, for instance, have shown no difference [51] while focal cortical dysplasias presented with increased overall connectivity [52,53].

The IMPC as a basis for node-degree calculation, as used in our study, has been applied before [39,54]. This method proved to be sensible at group levels but had poor within-subject repeatability compared to different approaches in a reliability study performed by Colclough et al. (2016) [33]. IMPC seemed to be less effective in shorter and noisier recordings than other network measures, which motivated us to use recordings of at least 4 min in length to improve the signal–noise ratio (SNR). Our results are a motivation to acquire more data with longer recording times of up to 20 min.

With ICC analysis showing a wide variability, longer recordings need to be performed to ascertain if increased SNR extract a more consistent basis of functional connectivity.

### 4.2. Influence of Vigilance

Previous studies have suggested functional connectivity to be stable over months and to be insensitive to fatigue or sleep deprivation [55,56,57]. Other studies found evidence for changes in fMRI connectivity within the hippocampal and thalamocortical system after 24 and 36 h of sleep deprivation [58,59]. Evidence on how connectivity or, to be even more specific, node degree is affected by short-term changes of vigilance, when fatigue emerges, could not be found in the current literature.

We did not observe systematic differences between measurements at the beginning and at the end of the recording session after a tiring task. However, ICC values were comparably low, ranging from −0.39 for gamma to 0.36 for theta. These results could have been falsified by interindividual differences regarding fatigue. While this shows very limited reliability, node degree values always preserved a significant contrast to the much higher node degree values of patients.

### 4.3. Delta

Furthermore, with ICCs of 0.53 and 0.36, reproducibility was best in the delta band in the eyes open/closed respectively Start/End conditions. This observation is reminiscent of findings on slow-wave (delta/theta) contrast between patients with epilepsy and healthy controls, which also extends to patients with recurrent seizures after epilepsy surgery vs. seizure-free patients [60]. Our observation regarding delta connectivity could thus be a correlation to the increased levels of such slow oscillations. It will be interesting to evaluate whether delta node degree also shows localizing potential, as has been demonstrated for slow-wave activity [61].

### 4.4. Beta

Both EC/EO and Start/End conditions showed the best contrast within the beta band and were significantly higher in the patient group. This is interesting, as beta activity is a rare ictal EEG pattern observed sometimes in specific subgroups, such as patients suffering from seizures due to A-V malformations, porencephalic cysts or tumors, in children with Lennox–Gastaut Syndrome, or infantile spasms [62], i.e., showing little overlap with our sample.

In addition, however, Heers et al. [63] have successfully utilized interictal beta-band activity to localize epileptogenic focal cortical dysplasias, demonstrating that increased beta activity related to epilepsy may occur interictally. Wang and Meng [64] have investigated network alterations in epilepsy and also report significant increases in some connectivity measures in the beta band, although node degree was not investigated.

Especially in the case of comparisons between patients and healthy controls, anti-seizure and other medication is a potential confounder regarding beta-band oscillations. For example, benzodiazepines have long been known to induce beta waves in the EEG [65], and Park and Kwon [66] demonstrate the induction of high beta oscillations by levetiracetam adjunctive therapy. Further work is thus required to disentangle medication effects from genuine factors associated with epileptic activity.

### 4.5. Expected Alpha/Low Gamma Alterations in Open vs. Closed Eyes

None of our data shows systematic differences in the EC vs. EO condition. We expected to see differences in the alpha band, as we assumed node degree might follow the behavior of oscillatory power in EO/EC conditions [67,68,69]. However, no increase of occipital alpha node degree in the EC condition could be observed.

The comparison of EO low gamma to the patient group did not show a significant contrast (*p* = 0.074), unlike any of the other conditions and frequency bands.

Effects regarding gamma have been found in other studies, suggesting EO connectivity to be more consistent when compared to EC. For example, Englot et al. (2015) have shown EC gamma connectivity retest reliability to be the least consistent [17].

The limited reliability of the EC condition is supported by Agcaoglu et al. (2019) who showed a wide variability of brain activity when tested with closed eyes because subjects are experiencing “mind-wandering” effects and concluded future measurements with open eyes to be more reliable [70].

Finally, the missing alpha node degree increase, and the higher gamma variability, may be due to the limited recording time. It seems conceivable that longer recordings might show a stronger influence on EC vs. EO in the alpha and potentially gamma bands. Comparisons with patients should therefore take EO/EC into account.

### 4.6. Power Comparison

As investigated in previous studies by our department, patients with focal epilepsy are thought to have increased oscillatory power in certain frequency bands [48,60]. Even though the chosen calculation method, IMPC, is not affected by signal amplitude [54,71], this increased power could have an impact on overall node degree and falsify our results.

To evaluate if there is an influence from increased oscillatory power or altered SNR, we added a power analysis comparing ND in controls to patients (Table 5/Figure 3). Alpha, beta and low gamma bands showed no significant power increase, while delta marginally missed significance (*p* = 0.074). The power level of theta reached statistical significance (*p* = 0.014).

However, contrary to our hypothesis of increased power levels in patients, both delta and theta power were decreased in patients, with corresponding AUC values < 0.5. We, therefore, suggest that nodal degree IMPC is robust against changes in oscillatory power.

### 4.7. Limitations and Outlook

Our study has several limitations. Firstly, the intake of antiepileptic drugs (AED) has been proven to influence EEG/MEG data, e.g., the decrease of gamma-band activity [72,73,74], as shown by Clemens et al. (2008) and Arzy et al. (2010). Yet, not much is known about its effect on connectivity or node degree. Consequently, there is a possible effect of AED on our data. To validate those findings, a future study with untreated patients or with a study design estimating the impact of dosage changes could fill these gaps.

Another point is the spatial leakage of the IMPC calculation [33], which could be causing errors within our data. However, spatial leakage would primarily affect localization accuracy, which was not the focus of our study. Data from healthy controls should be affected in a similar manner. Consequently, we believe that this quantitative contrast is less likely to be affected by this issue. As a follow-up, the results could be validated using longer recordings of 15 min or more. Additionally, simulation studies could investigate the effect on node degree.

Further elaborating this aspect, evaluating different connectivity measures as a basis for graph analysis could be interesting. While the IMPC already seems to adequately depict increased node degree in patients, specially directed measures, such as Granger causality [75], might add to this and provide insights into the propagation of interictal epileptic activity, as demonstrated in a recent study [76]. To evaluate propagation, a comparison with structural connectivity might also be interesting.

The sample size of this study was limited to 15 patients and 15 controls. Due to the unclear dimension of the investigated results, i.e., the patient-control contrast and the different experimental conditions, we did not perform sample-size calculations. However, our current results now provide the necessary data for the estimation of adequate sample sizes. They also imply that node degree may be a means of further investigating pathological activity and network changes in patients with epilepsy. Already, the quantitative differences to healthy controls may provide tools to evaluate and monitor the success of, for instance, pharmacological therapy and post-surgical recurrent seizures [60]. Arguably, however, the most interesting application would be its use for focus localizations. To this end, a study specifically investigating the distribution of node degree in relation to the epileptic focus confirmed by, e.g., invasive recordings or epilepsy surgery will be needed. This has not been the aim of the current study; it instead focused on node degree quantity in a representative consecutive patient series, avoiding selection bias as much as possible.

All our patient MEG data were recorded with EC as part of their presurgical workup. The results of the current study show that differing measurement conditions are not able to reproduce similarly high levels of node degree in healthy controls. However, the influence of different measurement conditions on node degree in patients is still unclear. It is thus conceivable that recording EO data or performing longer recordings to facilitate sleep might further improve the contrast, although the standard EC acquisitions seem already promising for practical applications.

The experimental setup is part of our limitations since we had a simple modulation of vigilance without further monitoring. Patients were asked if they fell asleep during recordings, which none of them affirmed, but future experiments might classify the level of vigilance by adding EEG surveillance for better categorization.

Epochs containing IEDs were not excluded from further analysis, as we considered such discharges as one example of putative spontaneous activity that is communicated to distant network nodes, giving rise to the observed node-degree changes. However, an increase in node degree despite low IED rates in our data, and a correspondingly non-significant correlation between these parameters, argue that IED do not constitute the entirety of such activity. Further research is required to determine the relationship between IED and connectivity, in regard to both quantity and localization.

## 5. Conclusions

To conclude, missing systematic differences within different experimental conditions demonstrate the potential of functional connectivity in MEG as an addition in diagnostics. Future studies with a greater population and longer recording time might address other connectivity measuring methods, as well as further node-degree-based source localization to evaluate and establish node degree connectivity as a possible presurgical marker.

## Figures and Tables

**Figure 1 brainsci-11-01590-f001:**
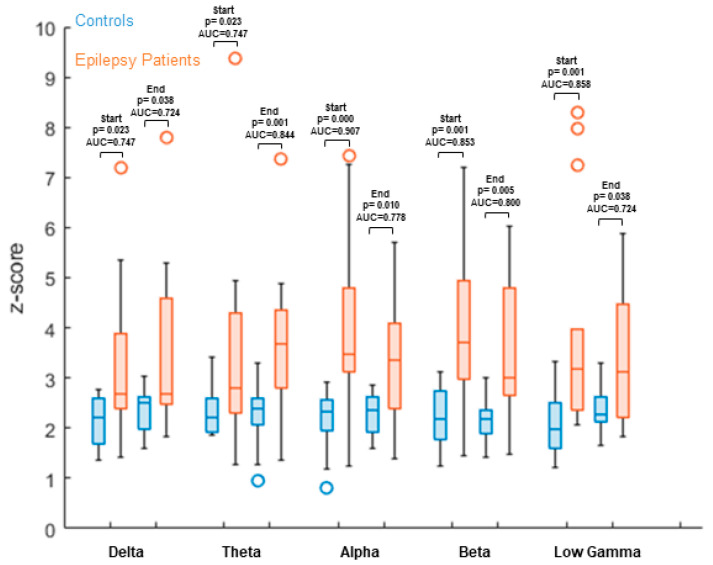
Max. node degree comparison between 15 healthy controls and 15 patients with focal epilepsy, comparing: (1) controls recording before an exhausting task with patient data; and (2) after an exhausting task with patient data. Results showed significantly increased values, as seen in graph data. Z-values are calculated based on the mean and standard deviations of the control group for each parcel. The mean value over all parcels is then calculated and used as the result of the individual subject. These results are then displayed separately per group. Z-values between groups are compared using a non-parametric Wilcoxon rank-sum test, as well as ROC AUC.

**Figure 2 brainsci-11-01590-f002:**
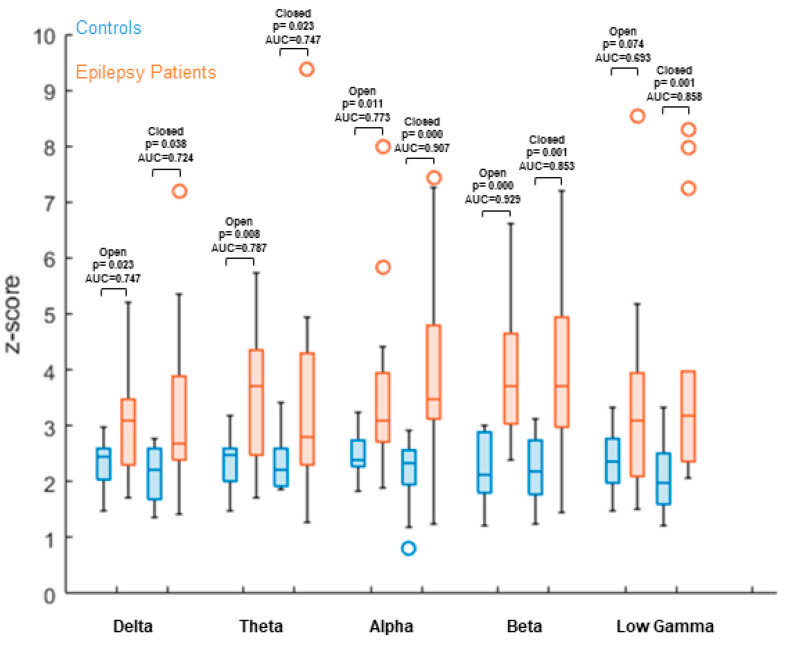
Max. node degree comparison between 15 healthy controls and 15 patients with focal epilepsy, comparing: (1) EO controls recording with EC patient data; and (2) EC control data with EC patient data. Results showed significantly increased values, as seen in the graph data. Z-values are calculated based on the mean and standard deviations of the control group, individually for each parcel. The mean value over all parcels is then calculated and used as the result for the individual subject. These results are then displayed separately per group. Z-values between groups are compared using a non-parametric Wilcoxon rank-sum test, as well as ROC AUC.

**Figure 3 brainsci-11-01590-f003:**
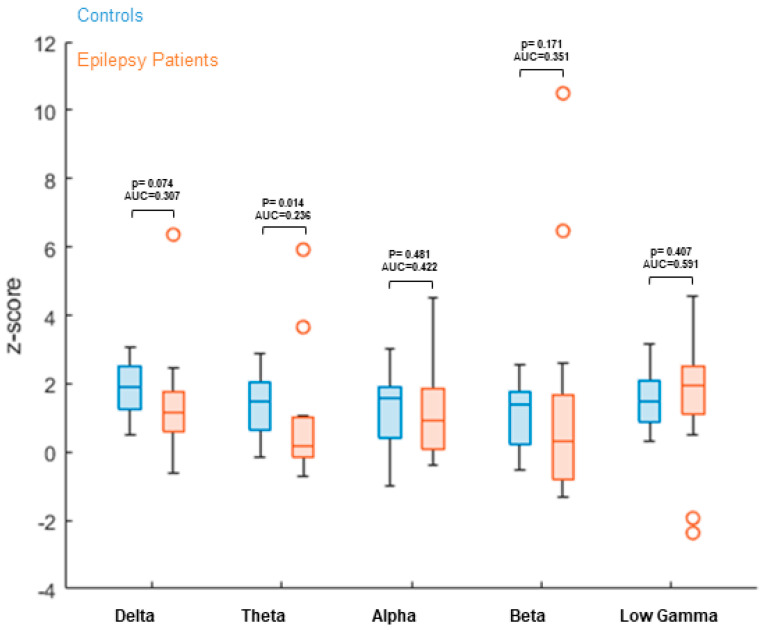
Power comparison between 15 healthy controls and 15 patients with focal epilepsy, comparing EC condition as it complies with our typical patient recording. Results showed no significant difference in ND power levels between patients and controls in alpha, beta and low gamma bands. Significant and close to significant results can be seen in delta and theta bands, with patient ND power lower than controls. Z-values are calculated based on the mean and standard deviation of the control group individually for each parcel. The mean value over all parcels is then calculated and used as the result of the individual subject. These results are then displayed separately per group. Z-values between groups are compared using a non-parametric Wilcoxon rank-sum test, as well as ROC AUC.

**Table 1 brainsci-11-01590-t001:** Patient characteristics.

No.	Age	Sex	Years with Epilepsy	Localization of Epilepsy	Spikes/ 40 min rec.	Etiology
1	34	f	35	left fronto-temporal lobe	32	lesional
2	44	f	11	right temporal lobe	9	lesional
3	25	f	16	right fronto-parietal lobe	0	non-lesional
4	46	f	35	left parieto-occipital lobe	149	non-lesional
5	60	f	48	left temporal lobe	14	non-lesional
6	29	m	3	left temporal lobe	10	non-lesional
7	23	m	23	right centro-cingular	>300	lesional
8	31	f	20	right frontal lobe	24	non-lesional
9	38	f	29	left hemisphere	0	non-lesional
10	50	m	15	temporal bilateral	16	non-lesional
11	24	m	15	left hemisphere	18	non-lesional
12	24	m	6	right temporal lobe	3	non-lesional
13	25	m	16	right temporal lobe	23	non-lesional
14	34	f	17	left temporal lobe	0	non-lesional
15	23	f	7	right hemisphere	22	non-lesional

**Table 2 brainsci-11-01590-t002:** Maximum node degree ICC (Max. ND ICC).

Frequency Bands	Controls Open/Closed	Controls Start/End
Delta	0.53	0.36
Theta	0.08	0.15
Alpha	0.29	0.13
Beta	−0.21	0.13
Low Gamma	0.48	−0.39

**Table 3 brainsci-11-01590-t003:** AUC with “Start” and “End” conditions, compared to patients.

Frequency Band	Delta	Theta	Alpha	Beta	Low Gamma
**Start**	*p* = 0.023; AUC=0.747	*p* = 0.023; AUC = 0.747	*p* = 0.000; AUC = 0.907	*p* = 0.001; AUC=0.853	*p* = 0.001; AUC = 0.858
**End**	*p* = 0.038; AUC = 0.724	*p* = 0.001; AUC = 0.844	*p* = 0.010; AUC = 0.778	*p* = 0.005; AUC = 0.800	*p* = 0.038; AUC = 0.724

(Note: Conditions “Start” and “End” correspond to before and after performing a fatigue-causing task).

**Table 4 brainsci-11-01590-t004:** AUC with conditions EO/EC compared to patients.

Frequency Band	Delta	Theta	Alpha	Beta	Low Gamma
**Open Eyes**	*p* = 0.038; AUC = 0.724	*p* = 0.008; AUC = 0.787	*p* = 0.011; AUC = 0.773	*p* = 0.000; AUC = 0.929	*p* = 0.074; AUC = 0.693
**Closed Eyes**	*p* = 0.023; AUC = 0.747	*p* = 0.023; AUC = 0.747	*p* = 0.000; AUC = 0.907	*p* = 0.001; AUC = 0.853	*p* = 0.001; AUC = 0.858

**Table 5 brainsci-11-01590-t005:** Power and AUC comparing patients—controls at EC condition (values not corrected for multiple comparisons).

Frequency Band	Delta	Theta	Alpha	Beta	Low Gamma
**EC Power/AUC**	*p* = 0.074; AUC = 0.307	*p* = 0.014; AUC = 0.236	*p* = 0.481; AUC = 0.422	*p* = 0.171; AUC = 0.351	*p* = 0.407; AUC = 0.591

## Data Availability

The data presented in this study are available on request from the corresponding author.

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
