# Peer review of "MEG Node Degree Differences in Patients with Focal Epilepsy vs. Controls—Influence of Experimental Conditions"

_brainsci, 2021, doi:10.3390/brainsci11121590_

Round 1

Reviewer 1 Report

In this piece of work, Vogel and colleagues examined differences in functional connectivity (i.e., node degree) assessed with MEG between patients with focal epilepsy and healthy controls. They calculated node degree at four frequency bands using a whole-brain approach based on regional parcellation. They observed higher overall node degree in patients with epilepsy compared to healthy controls. They also assessed the reproducibility of their findings in different conditions (e.g., eyes open, eyes closed, etc). Their findings add to ongoing research that examines functional connectivity in patients with epilepsy as a potential biomarker of epilepsy, which can serve as a possible presurgical marker. 

The topic is interesting, the manuscript is well-written, and the followed methods are robust. I have a few comments to be considered by the authors:

  1. It is unclear to me why the betta frequency band has been skipped in this analysis. I fully understand that there is a possible overlap of beta frequencies with muscular activities, but the same applies to the other examined frequency bands (e.g., low gamma). I would appreciate it if the authors can present the findings in the betta band too, or at least explain why they did not include this band in their analysis.
  2. Similar to my previous comment, the selection of frequency bands seems a little bit odd. Why low gamma has been selected between 25 and 40 Hz? Typically, the betta band reaches up to 30 Hz, and gamma extends upwards. 
  3. Several recent studies have shown evidence that MEG and high-density provide comparable localization findings in terms of presurgical evaluation of epilepsy patients. Although the focus of this study is MEG, I would appreciate a discussion in terms of both MEG and high-density EEG on how these two different techniques can complete each other in the assessment of the functional connectivity in the epileptic brain.  
  4. Table 1: Please remove patients who did not include in the study. I would also change the m/w to male/female in the sex/gender section.    

Author Response

Dear Reviewer,

Thank you very much for your contribution!

We've adressed your comments point-by-point in the attached PDF file.

Regards,

Stephan Vogel

Reviewer 2 Report

The general theoretical framework of the paper can be considered interesting, and the paper itself is fluent and well written.  Please refer to the following suggestions.

1) One of the main claim of the paper is that patients with epilepsy display increased nodal degree. Why it is interesting? A lot of work already showed changes in degree or other graph measures. Where the degree difference  is maximally expressed? For example it could be mainly expressed in the nodes including the epileptogenic, irritative or propagation zones. That could be interesting. But considering the heterogeneous clinical group, this could not be investigated. The supplementary images do not show this.

Second, the value of the connectivity could differ between the two groups because the basal oscillatory activity level is simply different. Did you compare the spectral power between the two group? I think it will be different, and of course this would explain the results at the connectivity level. Specially considering that you used coherence, which is highly comparable to a cross-correlation of the band power of two signals. Therefore claiming that the connectivity and consequently the degree differs across the two groups would have been a mere consequence of the power level. If no inferences were made explaining why these difference emerge between the two groups the additional value of using connectivity is lost.

In order to improve the work I would suggest to expand the clinical group, including only temporal or frontal. Effective functional connectivity measures like granger causality, phase transfer entropy or mutual information, can be used to have a directionality of what happens in target nodes or across all the nodes. Then you can relate this functional results with the structural connectivity, so that you can investigate for example if the epileptic nodes show higher out-degree, and  if this degree is related to and increase in the number of streams passing through those nodes. That could be of particular interest for the community. But at the moment I don't think that this paper represent a valuable contribution in the field.

2) No differences have been identified between pre and post task, so I should suppose that if I use I network based statistic approach no differences in the connectivity matrix should emerge. But, what type of task did you use? Which cognitive function was engaged? How did you measured the fatigue? This should have been better explained, with a specific hypothesis. None of this is present in this work at the moment.

3) No difference have been identified between open vs closed eyes. This is difficult to understand. There a lot of works focusing on the connectivity differences between the two conditions.

Considering that this work limited the investigation on the Node Degree without investigating its distribution across nodes or its' dynamic variability probably you did not find any difference in the NDD.   By concluding I would say that the work do not present methodological errors. However, stronger hypotheses and more detailed investigation is necessary to make these results interesting for the neuroscience and epilepsy community

Author Response

(The authors gave the same response as above.)
